# Spatiotemporal Dynamics of Ecosystem Services Driven by Human Modification over the Past Seven Decades: A Case Study of Sihu Agricultural Watershed, China

**Haowen Lin [1] and Hong Yun [2,*]**

[1]   College of Urban and Environment Sciences, Peking University, Beijing 100871, China
[2]   School of Design, South China University of Technology, Guangzhou 511436, China
*   Correspondence: yunhong90@scut.edu.cn

**Abstract:** Understanding the effects of human modification on ecosystem services is critical for effectively managing multiple services and achieving long-term sustainability. The historical dynamics of ecosystem services are important for detecting the impacts before and after intensive modification and deserve further study. To this end, we quantified the spatiotemporal dynamics of 11 ecosystem services across the Sihu agricultural watershed in 1954, 1983, 2001, and 2018. We used the Spearman coefficient, self-organized maps clustering, and redundancy analysis to explore the spatial patterns and potential modification drivers of temporal variations of ecosystem service provision. The results revealed the following: (1) The spatial correlations among ecosystem services in a single year were inconsistent with the ecosystem service change associations during two-time steps. The snapshot correlations at one time led to misunderstandings (such as water yield and runoff control or soil carbon sequestration, and habitat quality changed direction from synergy to trade-off) and missed synergies (such as water purification and recreational potential); (2) Most ecosystem services could be synergetic in one bundle with multifunctionality before intensive modification, but later transformed to single or limited services dominated bundles, especially in lake-polder areas; (3) Lake reclamation and hydraulic infrastructure were the most significant modification indicators explaining the variation of ecosystem services (30.9% of variance explained by lake reclamation in 1954, 38.2% of variance explained by hydraulic infrastructure in 2018). Meanwhile, changes in dominant drivers also indicated the transition from lake-based ecosystem service supply to engineered service. An improved understanding of the spatiotemporal pattern of ecosystem services and the underlying human modification influence is vital for realizing the sustainability and multifunctionality of agricultural watershed.

**Keywords:** ecosystem services; ecosystem service bundles; human modification drivers; historical dynamics; lake reclamation; hydraulic infrastructure; water regulation; soil carbon sequestration; habitat quality

## 1. Introduction

Ecosystem services (ES) are benefits humans acquire from the ecosystem, including material goods and services, which are closely bound up with human well-being [1]. Hence, humans have designed a series of intervention measures to manage and change the ecosystem to improve efficiency and maximize the delivery of certain services at the expense of others [2,3]. Especially in agriculture watersheds, human modification has resulted in highly stressed ecosystems and the loss of their services [4,5]. For example, anthropogenic inputs (e.g., fertilizer, pesticide, irrigation, and drainage canals) and agricultural expansion (e.g., lake or wetland reclamation) increase the crop yield significantly during agricultural intensification while causing water pollution [6], soil carbon storage reduction [7], biodiversity decrease [8,9], and damages from flood events [10]. Toward this end, understanding

the influence of human modification on ES is the prerequisite and foundation for effectively managing ES and achieving long-term sustainability [11–13].

Different types of ES may respond differently to modification. Pairs of ES are linked when an ecosystem service varies in response to driving factors that change the supply of another ecosystem service. "Trade-off" and "Synergy" are used to indicate the relationship between ES in most cases. Generally, ES synergy is a positive ES relevance (i.e., both services are increasing or decreasing), whereas ES trade-off is a negative relevance (i.e., one service is decreasing whereas another one is increasing) [14,15]. The trade-offs between food production and other regulating services are well documented in agricultural watersheds [16–18]. The trade-off analysis between ecosystem services is essential for detecting the diverse effects of modification [19]. The concept of an ES bundle, which is a set of associated ES repeatedly occurring together across time and space, provides an effective way to capture the impacts of modification on group ES [20]. Many methods have been developed to bundle ES, including k-means clustering [20–22], principal component analysis with k-means clustering [23,24], self-organizing maps (SOM) [25–27], and so on [28,29]. Compared with other approaches, SOM is a novel way to produce ES bundling, with the advantages of spatial clustering visualization and robustness in topology identification [19,30]. The bundling of ES permits the detection of critical patterns in the supply of ES driven by past drivers' dynamics [20], and the changing spatial clustering can inform the spatial planning and priority setting [31]. Therefore, exploring the synergy and trade-off through bundles can provide further insights into the impacts of modification on ES.

Although the assessment and mapping of ES and their bundles across different scales have been developed and emphasized in the past decade [29,32–36], little attention has been focused on the temporal variability and dynamic of ES [37,38]. Most previous studies provide snapshots of ES interactions in time, however, ES supply can strongly vary over time [39]. Some have argued that spatial correlations do not predict changes in ES [40,41]. The lack of historical dynamics in current ES evaluation may lead to misunderstandings [42,43]. For one, ES relationships, which are not static, may disappear or invert over time [44,45]. Even the well-known trade-off between agricultural production and water quality can transform into synergy with management interventions [46]. For another, drivers of ES change over time [47]. A few studies that have tried to quantify spatiotemporal patterns of ES are restricted in temporal scope, with a span of 20–30 years or limited ES categories [22,27,48]. Consequently, the historical ecosystem baseline involved in ES assessment is critical for managing landscapes for multiple services [14]. Additionally, temporal ES have important implications for comparing ecosystem management options before and after human modification [49].

To bridge the gap, this research aims to understand how ES vary through time and space in a typical agricultural watershed in the middle of Yangtze River Basin, China. We use historical primary and modeled ES data from 112 street districts across the Sihu agricultural watershed to quantify the spatiotemporal dynamics of 11 ES in 1954, 1983, 2001, and 2018. The specific objectives are: (1) to assess the spatiotemporal relationship between pairs of ES indicators; (2) to delineate ES bundles across time and space through SOM; and (3) to explore the effect of potential human modification drivers on ES.

## 2. Materials and Methods

### 2.1. Research Site

The Sihu agricultural watershed, covering a total area of 11,547.5 km², located at the hinterland of Jianghan Plain, borders the Yangtze River to the south, the Hanjiang River and Dongjing River to the north, and the Zhanghe Reservoir to the northwest, approaching the east longitude from 112°00′ to 114°00′ and the north latitude from 29°21′ to 30°29′ (Figure 1). The Sihu basin is named after four large lakes (Changhu Lake, Sanhu Lake, Bailu Lake, and Honghu Lake), and it is a main part of the Ancient Yun Meng Swamp. "The whole alluvial plain is a honeycomb of waterways and depressional areas encompassed by these waterways are dish-shaped in cross-section" [50]. The shallow lakes and marshes are typically located in the center of these depressions. Belonging to the humid area of subtropical

monsoon, the Sihu watershed features plenty of rain with a multi-year average precipitation of 1000–1350 mm, and the annual mean temperature is 15.7 °C–17 °C. Specifically, roughly 70% of its annual precipitation is concentrated in the flood season (from May to September). The terrain in the basin is flat, with plains and depressions accounting for 77.3% of the area. Its ground elevation generally ranges from 25 to 40 m [51].

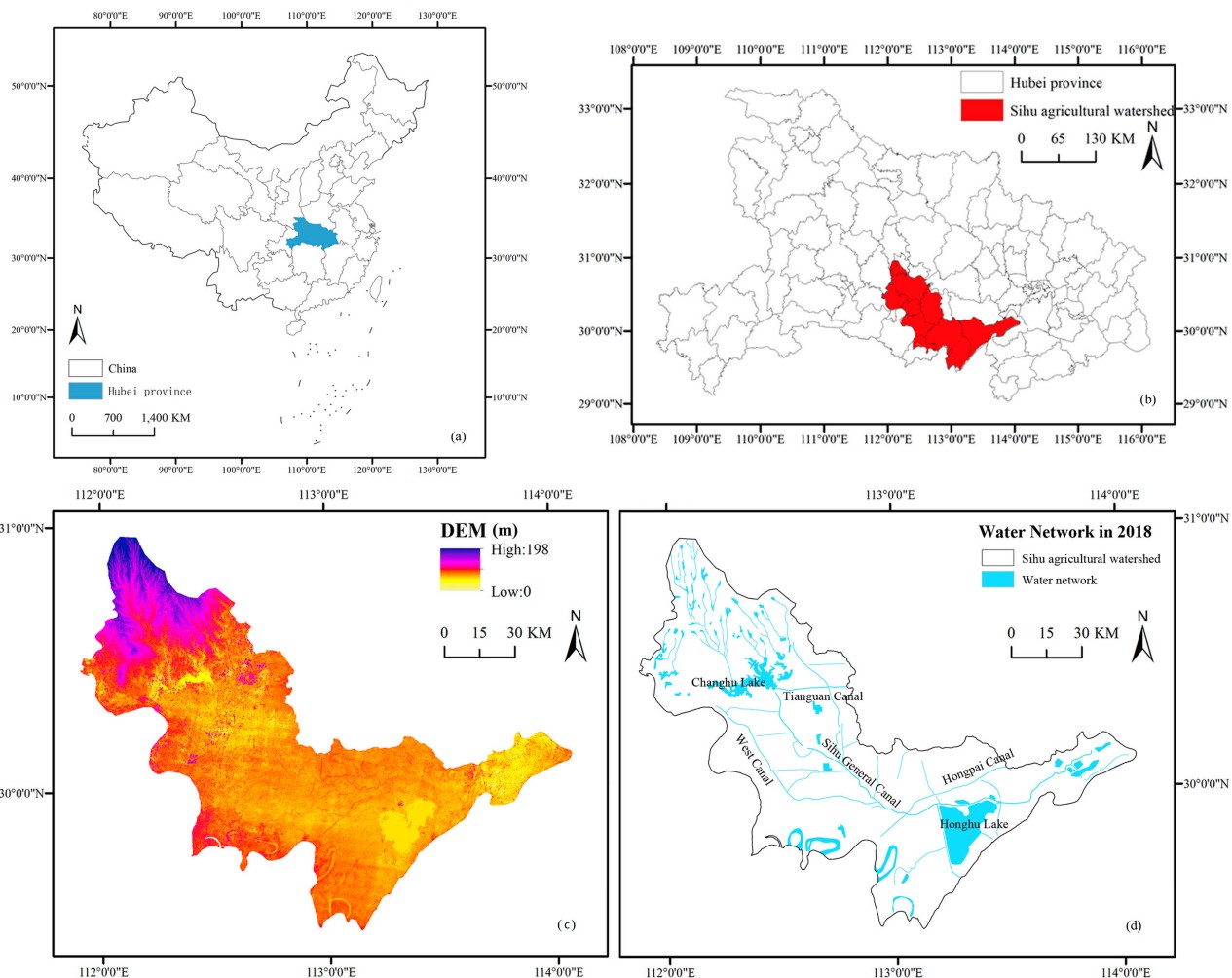

**Figure 1.** Geographic location, DEM and water network of the study area (**a**–**d**).

Large-scale water conservancy reconstruction has been launched for flood control, drainage, and irrigation since the Scheme for Flood Control and Drainage in Jingbei District was issued by the Changjiang Water Resources Commission in 1955. The Neijing River was dredged into the Sihu General Canal, followed by various water conservancy projects, such as large drainage channels, electric drainage pumping stations, flood control dikes, and irrigation culverts. The water level of the lake dropped rapidly after the building of Sihu General Canal, East Canal, and West Canal, which promoted the conversion from lake and wetland to agriculture. As a result, the area of the lake sharply decreased, losing nearly 2000 km² [52]. Reclaiming land from lakes has brought about serious ecosystem degradation issues, such as habitat loss and biodiversity decline, frequent floods, agricultural nonpoint-source pollution, and soil fertility decline, although it contributed to the development of Sihu into an agricultural basin rich in crop products [53]. Moreover, regional sustainable development has been seriously challenged by the strong trade-off between the ecosystem provision services and regulation services brought by the water conservancy development in the basin. Meanwhile, the transformation from a relatively

pristine, natural system to a hydraulic modified system provides a great opportunity to detect the effects of human modification on ES across spatiotemporal scales.

*2.2. Data Sources*

Five kinds of data are needed in ES assessment, including remote sensing image, land use, meteorology, soil, and demographic. (1) The topographic map of 30 m × 30 m was originated from the geospatial data cloud of the Chinese Academy of Sciences (http://www.gsclound.cn, accessed on 7 January 2023); (2) The land use data from 1983 to 2018 with a resolution of 30 m were originated from Landsat series of satellite remote sensing images and downloaded from the official website of the United States Geological Survey (https://www.usgs.gov, accessed on 7 January 2023) and the Geospatial Data Cloud (http://www.gsclound.cn, accessed on 7 January 2023). Five classes of land use types (settlement, lake, wetland, pond, and cropland) are interpreted based on supervised classification in Envi. The data from the 1950s were from the Four Lakes Water System Map (1954), the military aerial survey map (1959) provided by the archives of the Jingzhou Sihu Basin Engineering Management Bureau, and the Jingzhou Special Area Soil Distribution Map (1959) offered by the Agricultural Bureau of the Jingzhou Special Administration (Figure 2); (3) The daily meteorological data of meteorological stations in Jingmen, Qianjiang, Jingzhou, Jianli and Hong Lake from 1999 to 2019 were obtained from the China Meteorological Data Service Center (http://data.cma.cn, accessed on 7 January 2023), including precipitation, average wind speed, average temperature, sunshine hours, minimum temperature, maximum temperature, relative humidity and so on; (4) The soil data were downloaded from the Harmonized World Soil Database version 1.1 (HWSD) (IIASA—Land Use Change and Agriculture Program); (5) The demographic and social statistics were summarized in the statistical yearbooks of various administrative regions, covering the rural statistical yearbook of Hubei, the statistical yearbook of Jingzhou and the statistical yearbook of Jingmen (http://data.cnki.net, accessed on 7 January 2023).

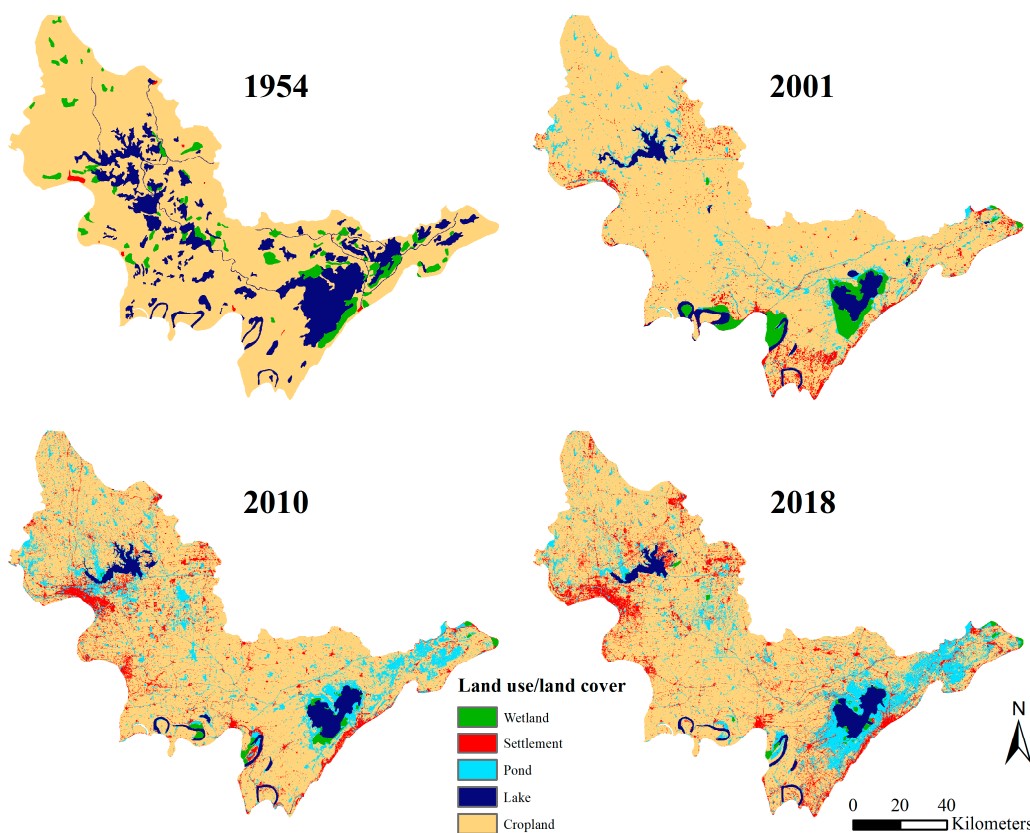

**Figure 2.** The land use/land cover in different years (1954, 1983, 2001, 2018).

*2.3. Mapping of Historical ES*

Thirteen indicators of eleven ES were selected according to the literature review and the site characteristics. These services were in line with the classification system, including Millennium Ecosystem Assessment [1], nature's contribution to people [54], and Common International Classification of Ecosystem Services framework version v.5.1 [55], by which ES were roughly divided into provision, regulation, and cultural services. Mixed indicators from potential capacity to actual supply were adopted in the study, as shown in Table 1 [24,41]. The annual supply of all ES in 1954, 1983, 2001, and 2018 was quantified with the street as the analysis unit. Most indicators were higher values corresponding to more ES supply, while runoff, total nitrogen output, and total phosphorus output were the opposite. All ES were evaluated at a resolution of 30 m, and then we calculated the average ES value of each street district in the Sihu basin. The provisioning ES were measured by the actual yield based on the statistical yearbook of different countries (Tables S1–S4). Different models were used to measure the potential ability of the ecosystem to deliver regulating and cultural services as follows.

**Table 1.** Overview of ES assessment in Sihu agriculture watershed.

| ES Category | Ecosystem Service | Indicators | Type | Unit |
| --- | --- | --- | --- | --- |
| Provisioning | Rice | Rice production | Actual | t/km$^2$·y |
| | Cotton | Cotton production | Actual | t/km$^2$·y |
| | Oil crop | Oil crop production | Actual | t/km$^2$·y |
| | Freshwater products | Freshwater products production | Actual | t/km$^2$·y |
| Regulating | Water yield | Water yield volume | Potential | mm |
| | Runoff control | Runoff volume | Potential | mm |
| | Habitat quality | Habitat quality index | Potential | Index (dimensionless) |
| | Water purification | TN, TP export | Potential | kg |
| | Carbon sequestration | Carbon storage capacity | Potential | t |
| | Soil quality | Soil organic matter, pH value | Potential | Index (dimensionless) |
| Cultural | Recreation | Recreation potential index | Potential | Index (dimensionless) |

2.3.1. Water Yield and Runoff Control

The ecosystem services of water yield and runoff control were evaluated through the Soil and Water Assessment Tool (SWAT). The SWAT is widely used to assess hydrologic or water-related ES [56–58]. The model is based on hydrological response units (HRUs). It includes two main parts during the whole simulation process. The surface process calculates the volume of runoff, while the river process simulates the transformation from reach to outlets [59]. The water balance equation used in the SWAT model is as follows:

$$SW_t = SW_0 + \sum_{i=1}^{t} \left( R_{day} - Q_{surf} - E_a - W_{seep} - Q_{gw} \right) \tag{1}$$

$$Q_{surf} = \frac{\left( R_{day} - 0.2S \right)^2}{R_{day} + 0.8S}; S = 25.4 \times \left( \frac{1000}{CN} - 10 \right) \tag{2}$$

$$WYLD = Q_{surf} + W_{seep} + Q_{gw} - TLOSS \tag{3}$$

where SW$_t$ is the final soil water content (mm), SW$_0$ is the initial soil water content (mm), R$_{day}$ is the precipitation, Q$_{surf}$ is the surface runoff, S is the potential maximum retention after runoff begins, CN is the curve number, E$_a$ is the evapotranspiration, W$_{seep}$ is the percolation and Q$_{gw}$ is the amount of return flow, WYLD is the water yield, TLOSS is the loss of reach due to riverbed leakage. The Q$_{surf}$ represents the ability of runoff control,

while the WYLD means the water yield services. The detail of model calibration and validation is given in Supplementary Materials Tables S5 and S6.

### 2.3.2. Habitat Quality

The habitat quality index is generated by the InVEST model. The habitat quality module combines the information of land cover and biodiversity threat factors [60]. The habitat quality module is composed of the relative impact of each threat, relative distance between grid unit and threat, as well as the relative sensitivity of each habitat type to each threat. Before assessing the impact of threat sources on habitat, it is assumed that the relative habitat suitability score of each land use type was from 0 to 1, with 1 representing the highest habitat suitability. The Honghu Lake was taken as a representative in the study due to the great differences in the habitat suitability of the same type in different years. The habitat suitability scores of wetlands and lakes in different years were modified using the biodiversity index (Supplementary Material S8). Then, the spatial attenuation degree of threat sources can be described through linear or exponential distance attenuation functions (Equations (4) and (5)). Next, the response of different habitat types to threats was modified by the introduction of sensitivity indicators (Equation (6)). Eventually, the degradation score was interpreted into the habitat quality score with the semi-saturation function (Equation (7)).

$$i_{\text{rxy}} = 1 - (\frac{d_{xy}}{d_{r\ max}}) \text{ if linear} \tag{4}$$

$$i_{\text{rxy}} = exp(-(\frac{2.99}{d_{r\ max}})\, d_{xy}) \text{ if exponential} \tag{5}$$

where $d_{xy}$ is the linear distance between pixel x and y; $d_{r\ max}$ is the maximum distance of threat r.

$$D_{\text{xj}} = \sum_{r=1}^{R} \sum_{y=1}^{Y_r} (\frac{W_r}{\sum_{r=1}^{R} W_r}) \times r_y \times i_{\text{rxy}} \times \beta_x \times S_{\text{jr}} \tag{6}$$

where R is the number of ecological threat factors, $W_r$ is the threat weight that relates destructiveness of a degradation source to all habitats, $Y_r$ is the set of pixels on threat r raster map, y is all the pixels on threat r raster map, $r_y$ is the threat r that originates in pixel y; $\beta_x$ is the level of accessibility in pixel x, $S_{\text{jr}}$ is the sensitivity of land use type j to the ecological threat factor r.

$$Q_{\text{xj}} = H_j \times (1 - (\frac{D_{xj}^z}{D_{xj}^z + k^z})) \tag{7}$$

where $H_j$ is the habitat quality score that ranges from 0 to 1; k is the half-saturation constant and was set at 0.05. The biophysical table for the habitat quality model is shown in Supplementary Materials Table S7.

### 2.3.3. Water Purification

The water quality purification service was measured by combining the Nutrients Delivery Ratio (NDR) module of InVEST with the lake monitoring data. The NDR model represented the long-term stable flow of nutrients through an empirical relationship, instead of the details of nutrient circulation [60]. The output results were highly sensitive to the small number of model parameters [61]. The goal of the model is to provide relative spatial comparison under different land use changes instead of predicting the nutrient output accurately [62]. Therefore, only a part of the output results of modified nutrient load were used. Then, the nitrogen and phosphorus exports of the basin were calculated with the watershed nutrient load output by the model and the observed nitrogen and phosphorus retention efficiency of the lake.

$$modified.load_{x_i} = load_{x_i} \times RPI_{x_i} \tag{8}$$

$$RPI_i = RP_i / RP_{av} \tag{9}$$

$$Export_{TN/TP} = modified.load_{x_i} - modified.load_{x_i} \times LR \tag{10}$$

where $RPI_i$ is the runoff potential index on pixel i, $RP_i$ is the nutrient runoff proxy for runoff on pixel i, $RP_{av}$ is the average RP over the raster, the raster RP is defined as precipitation, $Export_{TN/TP}$ is the total watershed export of TN or TP (kg), $load_{x_i}$ is the nutrients load of each land use type on pixel i, LR is the nutrients retention efficiency of lake. The calculation of the TN and TP load focuses on the agricultural non-point source pollution, including crop planting pollution, livestock breeding pollution, aquaculture pollution, and rural life pollution. The details of $load_{x_i}$ processing are presented in Supplementary Materials Tables S9–S11. Based on the monitoring data of Honghu lake published by Jingzhou Municipal Ecological Environment Bureau during 2016–2018 (http://sthjj.jingzhou.gov.cn/, accessed on 10 January 2023), the LR of TP and TN were set as 0.52 and 0.36. The biophysical table needed in the model is shown in Supplementary Materials Table S12.

### 2.3.4. Carbon Sequestration

Cropland is a main type of land use in the basin, and its carbon storage is the focus of research. Farmland ecosystems cannot store carbon for a long time due to the re-discharging of carbon absorbed by plants into the atmosphere in the form of $CO_2$ [63]. Therefore, instead of vegetation carbon storage, only soil carbon is calculated in the farmland ecosystem. Soil organic carbon density and soil carbon storage are two important concepts in carbon sequestration, which are calculated by the following Equations (11) and (12).

$$SOC_{density} = C \times \theta \times D \times (1 - \delta)/100 \tag{11}$$

where $SOC_{density}$ is the soil organic carbon density ($kg/m^2$), C is the average organic carbon abundance (g/kg), D is the soil depth and was set at 100 cm, $\theta$ is soil bulk density ($g/cm^3$), $\delta$ is the diameter of gravel content over 2 mm (%).

$$SOC_{storage} = S \times SOC_{density} \tag{12}$$

where $SOC_{storage}$ is the volume of soil organic carbon storage (kg), S is the area of cultivated land ($m^2$). The data of $SOC_{density}$ in 2018 were based on the research of Liu et al. and downloaded from National Earth System Data Center (http://www.geodata.cn, accessed on 10 January 2023). $SOC_{density}$ in 1983 and 2001 was generated by the ordinary kriging method based on the data of 112 soil profiles from soil chronicles. $SOC_{density}$ in 1954 was extracted from the soil map of Jingzhou in 1959 (Figure S2).

### 2.3.5. Soil Quality

The proxy of soil quality usually includes the indicators from soil physical, chemical, and biological properties. The concept of the minimum data set (MDS) of soil parameters was adopted [64], while considering the soil usage [65]. Therefore, the soil organic matter and pH value were used to measure the change of soil conditions for describing the long-term impact of farmland cultivation and fertilizer application on soil quality in the Sihu basin. Please refer to Supplementary Materials Figure S3 for the soil organic matter and pH value data of cultivated land in different decades.

### 2.3.6. Recreation

The recreational service in the Sihu basin was evaluated on the basis of the recreational scoring method by [18]. The natural proximity, biodiversity, cultural and historical characteristics are the three key indexes to assess the potential of recreation. The scores of three indexes are added according to the following Equation (13)

$$RP_i = \sum_{i=1}^{n}(SN_i + SB_i + SH_i) \tag{13}$$

where RP$_i$ is the total scores of recreation potential, SN$_i$ is the scores of natural proximity, SB$_i$ is the scores of biodiversity and was set as the habitat quality index, SH$_i$ is the scores of historical characteristics. The SN indicated the distance of each pixel from the nearest water, which was analyzed through Euclidean distance in GIS. The historical characteristics included the numbers of tourist attractions, temples, parks and plazas, which were obtained from the POI data and local toponymy chronicles. Then Euclidean Distance was conducted to calculate the nearest distance between each pixel and historical characteristics attractions.

### 2.4. Analysis of the Relationship between ES

Correlation analysis is used to identify trade-off or synergy between two ES [41,66]. The study not only calculated the spatial correlation in 1954 and 2018, but also counted the variation of each ecosystem service between two consecutive time periods (1954–1983, 1983–2001, 2001–2018) and the variation between the beginning and end of the entire time period (1954–2018). In addition, the relationship between the changed ES was calculated using the Spearman correlation coefficient. This way of analyzing intergenerational trade-offs of ES in different periods is widely used in the literature [24,67]. The analysis was conducted using ggplot2 package in R (version 4.2.2).

### 2.5. Analysis of ES Bundles

ES bundles refer to a series of combinations of multiple types of ES appearing repeatedly in space and time, which applies to the research on the relationship between multiple ecosystem services [20]. SOM was applied to reduce data dimensions and achieve the recognition and partition of ecosystem service bundles via clustering [26,30]. SOM was implemented through calling the Kohonen R package. By first setting the clustering range at 2 to 20, the minimum Davies–Bouldin index was acquired using the SOM method. Then, the Davies–Bouldin index in the ClusterSim R package could be leveraged to determine the most appropriate number of bundles [68]. Finally, flower graphs of the averages of various ES in each ES bundle were plotted via the ggplot2 R package.

### 2.6. Analysis on the Drivers of ES

The driver is the factor causing the change in the ES relationship. The impact of human modification drivers on ES is reflected by selecting population density, hydraulic engineering intensity, lake reclamation intensity, consumption of fertilizer application, and consumption of insecticide application. These indexes were extracted from street scale data and the relevant literature. Data preparation for potential drivers is described in the Supplementary Materials Table S13. Canonical analysis can well describe the causality between multiple ES and potential drivers. The relationship between the ES and drivers was analyzed using redundancy analysis in Canoco 5, which also obtained the explanation degree of drivers to the differences in ES.

## 3. Results

### 3.1. Temporal Patterns of ES Snapshot Correlations and Their Change Correlations

The Figure 3 showed the spatiotemporal pattern of individual ES in 1954, 1983, 2001, 2018. The snapshot correlations in 1954 and 2018 shared the same trend in provision services. There was a strong positive relationship between rice, cotton, and oil crops, but a negative relationship between oil crops and freshwater products was presented for both years. The runoff control and water yield services had positive correlations with other regulating and recreational services, whose relationships became more and more stronger from 1954 to 2018. There was a negative correlation between habitat quality and total phosphorus and nitrogen export in 1954; however, the relationships turned to positive in 2018. Compared to 1954, many significant positive correlations were fostered between soil carbon storage and other services in 2018, such as carbon storage and habitat quality, and carbon storage and recreation.

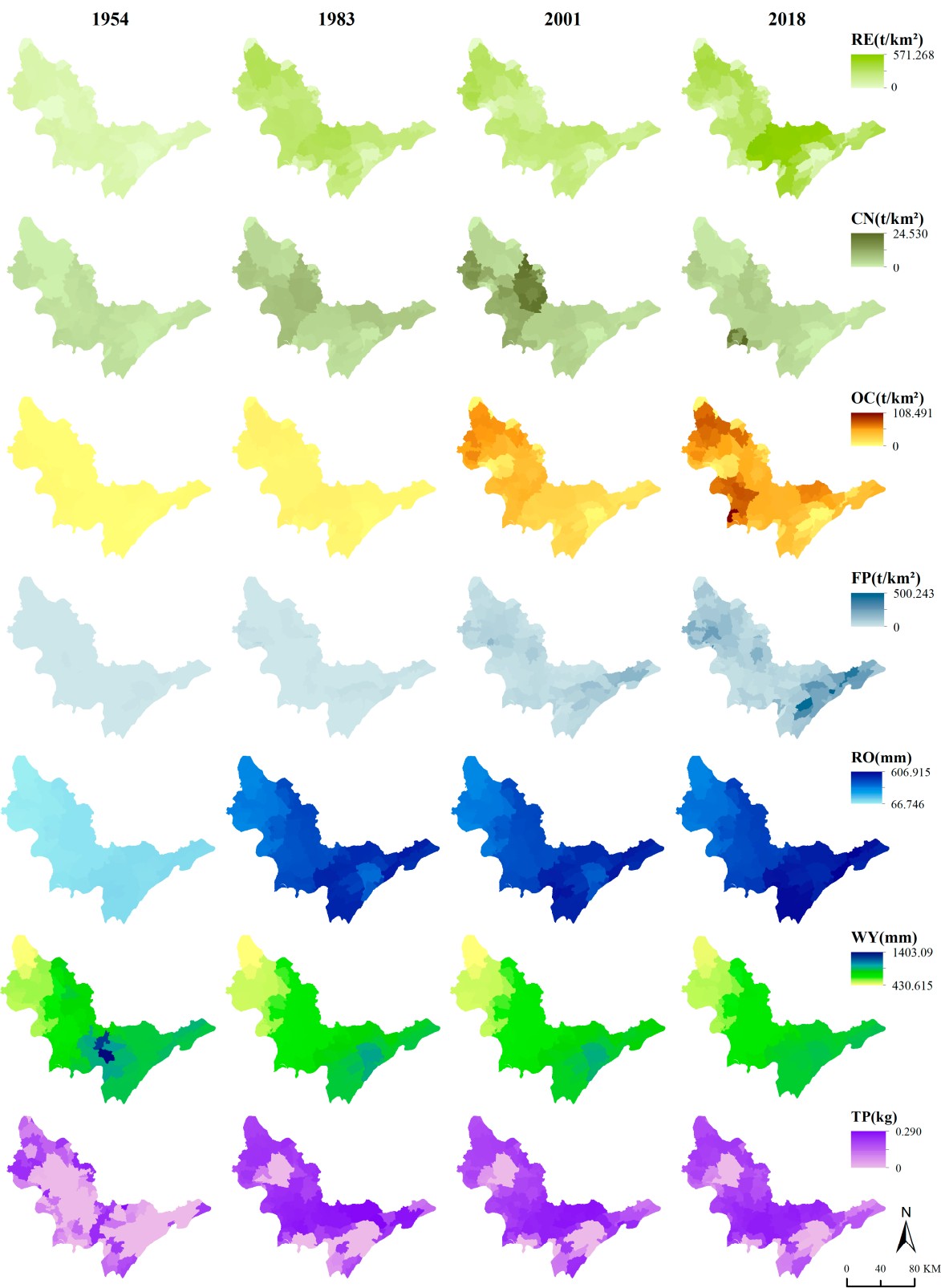

**Figure 3.** *Cont.*

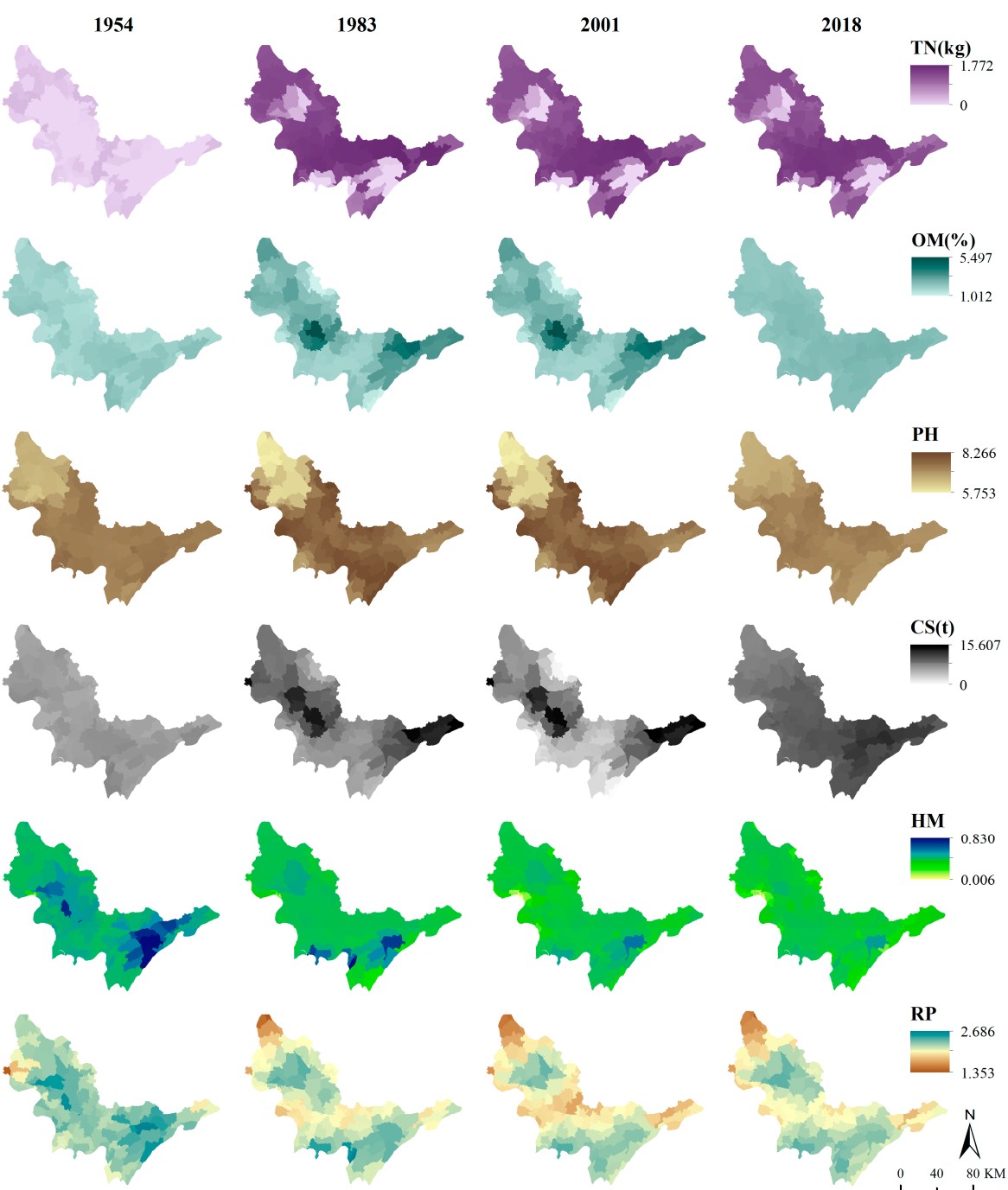

**Figure 3.** The spatiotemporal pattern of individual ES in 1954, 1983, 2001, 2018. RE: rice; CN: cotton; OC: oil crop; FP: freshwater products; WY: water yield; RO: runoff; HQ: habitat quality; TP: total phosphorus export; TN: total nitrogen export; CS: carbon storage; OM: soil organic matter; pH: pH value; RP: recreation potential. Low to high indicates the values of ES.

Various positive and negative relationships were observed over time from the intergenerational relationship of ES (Figure 4). A positive correlation means that a pair of ES evolve in the same direction. In contrast, a negative correlation presents that a pair of ES changes in opposite directions. Provision services such as food, cotton, and oil maintain a synergistic relationship in different periods. Only aquatic product supply has a certain degree of trade-off with other supplies. Water purification services in the basin generally decrease with rising grain products. In the past two decades, a trade-off was shown in the

supply of aquatic products, with various regulating and cultural services, especially soil organic matter and habitat quality.

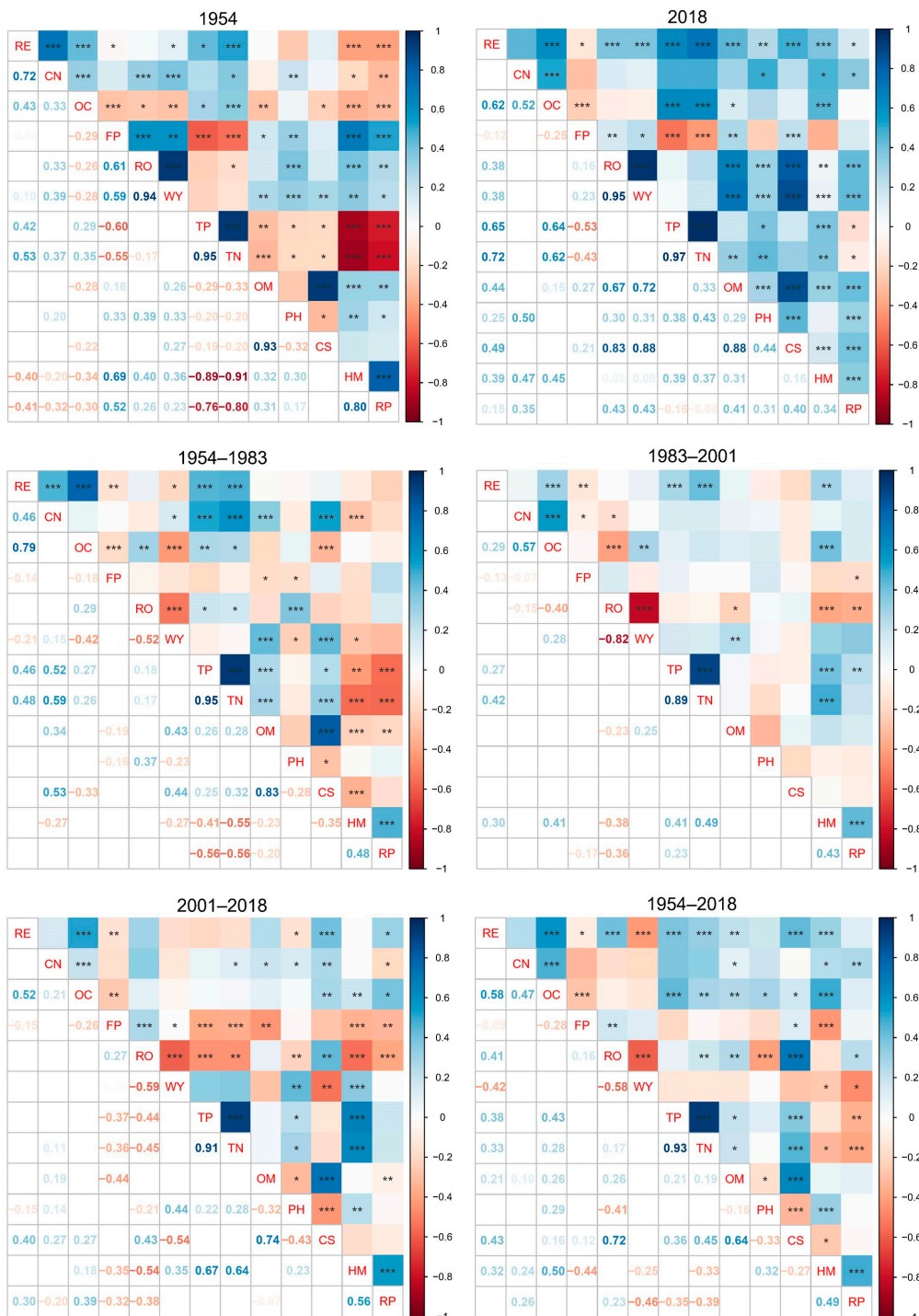

**Figure 4.** Snapshot correlations (1954, 2018) and change correlations (1954–1983, 1983–2001, 2001–2018, 1954–2018) between pairs of ES in Sihu agricultural watershed. The blue and red squares show the positive and negative correlations between ES. The asterisks in the squares show the significance degree (*** for *p* < 0.001, ** for *p* < 0.01, * for *p* < 0.05). The numbers indicate the Spearman's correlation coefficients. RE: rice; CN: cotton; OC: oil crop; FP: freshwater products; WY: water yield; RO: runoff; HQ: habitat quality; TP: total phosphorus export; TN: total nitrogen export; CS: carbon storage; OM: soil organic matter; pH: pH value; RP: recreation potential.

The intergenerational relationship of regulating services varied greatly over time. The changes of runoff control maintained high consistency with water yield in different time periods. The water yield had a downward trend with the increasing runoff. Meanwhile, there was always a highly positive correlation between total outputs of nitrogen and phosphorus and a negative correlation between the pH value and carbon sequestration. Changes in total nitrogen and total phosphorus in 1954–1983 and 1954–2018 were contrary to those of habitat quality and recreational services. Conversely, the other two periods kept their synergies. The rising organic matter was accompanied by increasing soil carbon sequestration in most cases. Cultural service had varying trade-offs with other services, except for its synergy with habitat quality.

The spatial correlations changed direction when the corresponding change correlation was considered. The snapshot correlations in 1954 with trade-offs moved towards synergies in 2018. However, this trend does not hold after considering the changes between the ages. For one thing, intergenerational ES had different degrees of trade-off and synergy, which were different from those of most ES, with a highly positive correlation in 2018. Taking the runoff and water yield as an example, the ES in 1954 and 2018 were synergistic in space and turned into a trade-off after adding the time dimension. The relationship between carbon storage and habitat quality also had the same trend through comparing the snapshot correlation in 2018 with the change correlation during 1954–1983 and 1954–2018. Additionally, the synergy of carbon storage and H value in 2018 completely changed direction into trade-off during 1954–1983, 2001–2018 and 1954–2018. For another, the formation of strong positive correlation among ES, such as recreation service and water purification service, or the disappearance of a synergistic correlation between ES, such as habitat quality and soil organic matter, were presented with the comparison between snapshot correlation and change correlation.

*3.2. Spatial-Temporal Patterns of ES Bundles*

ES in 1954, 2001, and 2018 were clustered into eight categories with the SOM-based clustering analysis (Figure 5). Bundle 3, bundle 5 and bundle 6 were inlaid and distributed in the middle of the basin in the 1950s. This region has provided abundant food, cotton and oil supply services, together with high regulation and cultural services. All kinds of services have achieved effective synergy, except that the water quality regulation service of bundle 2 was slightly lower. Bundle 4 and bundle 7, distributed around Honghu Lake, Datong Lake, and Changhu Lake, had the highest regulation and cultural services together with low supply services. Bundle 1, bundle 2 and bundle 8, located at the northwest of the basin, were dominated by a few ES, which featured high terrain and strong runoff regulation for their positions in the upper reaches of the river basin. Bundle 2 emphasized agricultural production, whereas bundle 8 had a high soil pH value.

By 2018, ES in the middle of the basin were dominated by bundle ③ and bundle ⑥, accounting for the largest area. Compared with ES in the 1950s, the water purification service vanished in the middle of the basin with the weakened synergy between various ES. Even worse, bundle ⑥ also showed an intensive trade-off between agricultural production and water quality regulation. Bundle ① and bundle ② in the upper reaches of the basin witnessed a declined average of other synergy services, although they were still dominated by runoff regulation services. Bundle ③ and bundle ⑦, dominated by Changhu Lake and Honghu Lake, were presented by similar clustering, presenting a clear trade-off among product supply service, regulation, and cultural service. Bundle ⑧ in the lower reach of the basin had poor runoff regulation capacity, although it had a high water yield.

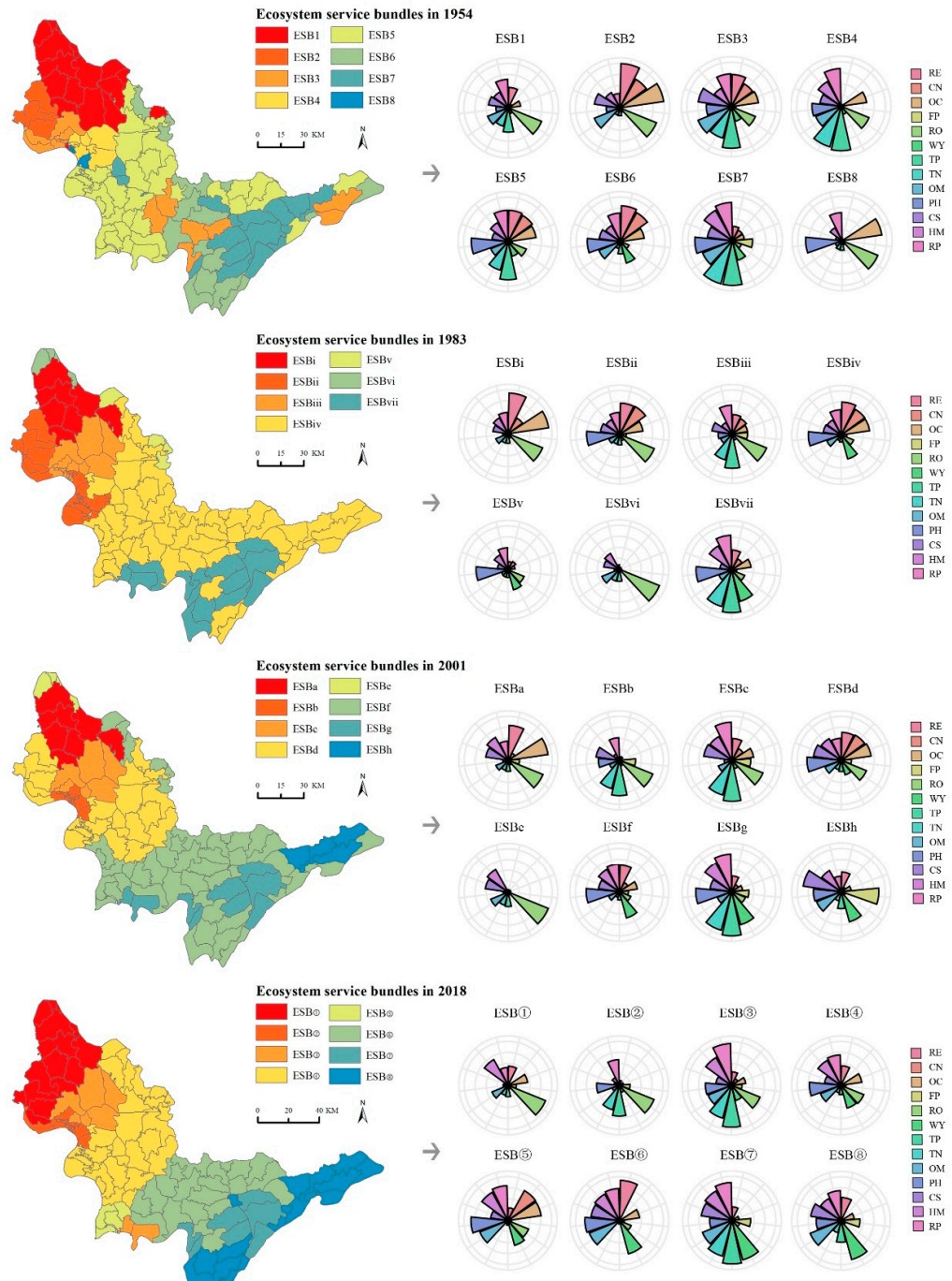

**Figure 5.** The ES bundles in different years and their corresponding flower graphs (1954, 1983, 2001, 2018). RE: rice; CN: cotton; OC: oil crop; FP: freshwater products; WY: water yield; RO: runoff; HQ: habitat quality; TP: total phosphorus export; TN: total nitrogen export; CS: carbon storage; OM: soil organic matter; pH: pH value; RP: recreation potential.

The spatial distribution of different bundles changed over time. The bundles in 1954 have a different pattern with other years, but the spatial characteristics of bundles remained relatively static after 1983. Ecosystem service bundles in 1954 were permeated in all areas of the basin, indicating the basins with diverse landscapes. During 2001–2018, ES bundles tended to present a clear division and change little. The greatest change of ES bundle distribution happened in the middle and lower reaches of watershed, which was exactly

the lake-polder area. These areas were contained by the agricultural bundles from 1983, while a different combination of agricultural products formed different bundles.

### 3.3. Relationships between ES and Human Modification Drivers

Redundancy analysis showed that human modification drivers were significantly correlated with changes in ES ($p < 0.05$) (Figure 6). Population density, hydraulic engineering intensity, lake reclamation intensity, fertilizer and insecticide input were driving forces that could explain 40% to 62.2% of the distribution and change of ES in the Sihu basin (Table 2). The degrees of various drivers explaining the difference in ES were varied over time. Reclaiming land from the lake was the core driving force of changes in ES in the early stage of basin development. Fertilizer input and lakes were joint influences by the 1980s. With the continuous enhancement of human modification, ecosystem services were impacted to a certain degree by various drivers under the leadership of water conservancy projects. The long time-span analysis also manifested that the intergenerational differences of ES were closely related to the changes of drivers.

**Table 2.** Redundancy analysis in a different single year or the change of two-time steps (1954, 1983, 2018, 1954–1983, 1954–2018).

| Human Modification Drivers | 1954 Percentage of Variance Explained (%) | 1983 Percentage of Variance Explained (%) | 2018 Percentage of Variance Explained (%) | 1954–1983 Percentage of Variance Explained (%) | 1954–2018 Percentage of Variance Explained (%) |
|---|---|---|---|---|---|
| All constrained | 42.7 | 47.9 | 62.2 | 40 | 41.3 |
| Lake reclamation intensity | 30.9 | 16.5 | 10.9 | 21.5 | 6.8 |
| Insecticide intensity | 9.8 | 2.5 | 8.8 | 2.3 | 6.6 |
| Population density | 2.1 | 3.9 | 1.9 | 3.9 | - |
| Fertilizer intensity | - | 20.1 | 2.4 | 11.7 | - |
| Hydraulic engineering intensity | - | 4.9 | 38.2 | 3.2 | 25.6 |

Great changes had taken place in the relationship between human modification drivers and ES. All kinds of regulation and cultural services were highly connected in 1954 through lakes, whose existence also greatly improved multiple services. The water purification, habitat quality, recreational potential and freshwater products were more relevant to lake reclamation intensity than water yield, runoff control, soil quality and carbon sequestration. In 1983, the fertilizer intensity had a strong positive correlation with agricultural production, organic matter and carbon storage. Still, the positive relationship between lake reclamation intensity and habitat quality and recreation remained, but the relevance declined. Until 2018, the hydraulic engineering intensity was positively related to soil quality, recreation, cotton production and carbon storage, but its relationships with water yield and runoff control were negative. Insecticide intensity was more related to rice and oil crop production. From the long-term change analysis, a positive correlation can be observed between lakes and services, such as habitat quality and recreation potential. With the strengthened hydraulic engineering, the water purification capacity was lowered, while runoff increased and carbon storage was enhanced. The explanatory power of the population density driver for ES was limited across the whole time.

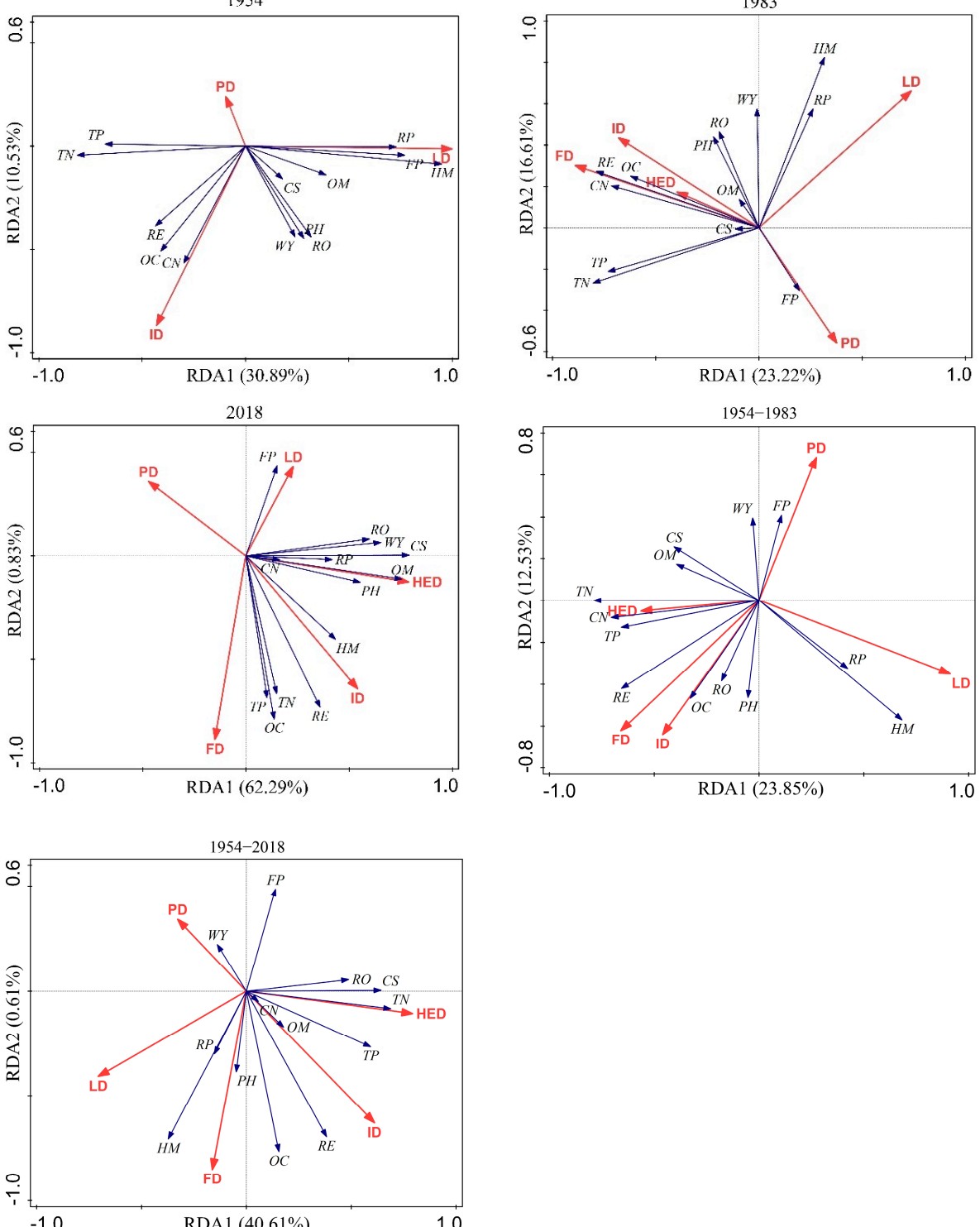

**Figure 6.** Redundancy analysis of ES (blue) and human modification drivers (red) in a different single year or the change of two-time steps (1954, 1983, 2018, 1954–1983, 1954–2018). The angles between arrows represent the strengths of the correlations between ES and human modification drivers. RDA: redundancy analysis; RE: rice; CN: cotton; OC: oil crop; FP: freshwater products; WY: water yield; RO: runoff; HQ: habitat quality; TP: total phosphorus export; TN: total nitrogen export; CS: carbon storage; OM: soil organic matter; pH: pH value; RP: recreation potential; PD: population density; HED: hydraulic engineering intensity; LD: lake reclamation intensity; FD: fertilizer intensity; ID: insecticide intensity.

## 4. Discussion

### 4.1. The ES Interactions Misunderstood without Historical Baseline

The relationship between ecosystem services is varied over time [24,69]. In line with past studies [20,69,70], we observed strong negative snapshot correlations between agricultural production and other regulating ecosystem services but strong positive relationships between most regulating services in 1954. However, the relationship between food production and regulating services changed direction in 2018. When there was higher rice product output, the soil organic matter and carbon storage were richer. This could be explained by the time lag in the feedback of ecosystem processes [47,71]. During 1954–1983, the lake reclamation improved the soil ferity dramatically through the cultivation of lakes or marsh sediment. After that, the average soil organic carbon slowed down because of agricultural intensification, but it is still relatively higher than in the 1950s. Rice planting also tended to happen in places where soil organic matter is rich. Moreover, it was a surprise to find the positive relationship between habitat quality and crop production in 2018, which tends to contradict previous results [25,72]. One explanation could be the model setting, as the habitat quality index was not highly sensitive to the management measures but changed based on the land use proxy. During 2001–2018, a large scale of cultivated land was converted to fishponds for higher profit. Compared to settlements and ponds, the agricultural areas were certain to have more biodiversity. This was also the reason why habitat quality had strong positive correlations with total phosphorus and nitrogen exports.

Similar to past studies of historical ES [40,69,73], snapshot synergies and trade-offs between ES do not necessarily correspond to changes in ES through time. Two variations were found after the incorporation of time, as shown in the following: (1) some of the positive correlations changed to negative correlations, such as water yield and runoff; and (2) a new service synergy was presented, such as the change in recreation potential being significantly positively correlated with the basin water purification service. There were two main causes. First, the landscape function changed dramatically. Similar landscapes could deliver different ES [74]. For instance, the increasing water yield generally provides more runoffs. However, the large number of lakes in the basin in 1954 effectively intercepted runoff, with a role in runoff regulation. By 2018, the site played no role in regulation and storage, although it had a large area of water surfaces composed of aquaculture ponds. The ponds remained at a fixed water level all year round. In the flood season, waterlogging in ponds would be drained quickly to prevent its influence on aquaculture. Evidently, the limited regulation and storage space of these fishponds could cause the joint increase of water yield and runoff. This was also verified by the relationship between the intergenerational ES of water yield and runoff. The relationship between ES was affected by the change in water surface properties. Moreover, the functioning time of driving factors also varied [73]. The synergy between recreational potential and water purification services was not detected from the analysis of intergenerational ES in the ranges from 1983 to 2001 and from 2001 to 2018. The reason might be that lakes had been reclaimed and had vanished before 1983. Without the involvement of historical ES, the possibility to foster the synergies may be ignored. Overall, the analysis showed that selecting an appropriate reference system is essential for judging the impact of human modification on the interaction of ES.

### 4.2. Lake Mediates ES Bundles

Consistent with the previous studies on ecosystem service bundles [29,75–77], the detected agricultural bundle was characterized mostly by agriculture–freshwater products. Due to the intensification and specialization of agricultural production, the distribution of the agricultural bundle has increasingly expanded and concentrated since 1983 in the Sihu watershed, which was also happening in other regions [25,69]. Additionally, the settlement bundle contributed little to supply services, while the lake bundle provided high levels of regulating and cultural services, which supported the findings of past studies [72,73]. However, there was one exception in 1954: ecosystem services could be synergetic in one

bundle. In the middle and lower reaches of the river basin, the trade-offs among ES could be partly balanced to improve the joint provision of multiple ecosystem services [18]. One possible important reason was the existence of the lake (such as SanhuLake, Bailu Lake, Datong Lake, and Dasha Lake). The lake not only purified the pollutants from agricultural activities but also reduced the runoff volume through stormwater retention. At the same time, lakes provided a great opportunity for recreation. Therefore, the potential of one land cover type to provide many ecosystem services may be as important as landscape complexity in the establishment of multifunctionality, as previously demonstrated in mountain regions [32] and the Europe agricultural landscape [78].

Compared with the ES bundles in 1954, the changes in the distribution of bundles in other years were likely to occur relatively slowly. This could also be found in previous results [23,27]. On one hand, the bundles were primarily determined by broadscale patterns of the biophysical processes [40,79]. The change in the general social-ecological drivers underlying the ecosystem biophysical processes has remained relatively stable since 1983. On the other hand, this might be associated with the simplification of land functions and the homogeneity of landscapes. The bundles provided by each street district became more and more specialized and clustered to large clusters in space. The pattern of ES bundles in 1954 distinguished the important role of lakes in enriching the diversity of ES bundles across the whole watershed. In the human-modified agricultural landscape, lakes are essential for mediating the competing services.

*4.3. The Transition from Lake-Based ES Supply to Engineered Services*

The costs of providing ES through artificial engineering instead of through nature are required [11]. Meanwhile, long-term investment and maintenance that might lead to path-dependence shall be also considered [12,80]. Moreover, it is difficult to be completely aware of the unexpected disasters caused by such a replacement [81]. The redundancy analysis showed that the capability of lakes to explain the changes of ecosystem services in the basin declined with the increasingly prominent impact of hydraulic engineering projects. With enhanced human modification, the lake had a weakened link with ES. This could be explained by the land use conflict. With the aid of improved drainage conditions and increasing inputs of chemical fertilizer and pesticides, large areas of lakes were reclaimed as cropland, which resulted in negative ecological consequences, such as frequent flooding, decline of biodiversity, water pollution and the extinction of some endemic species [13,82,83]. This was consistent with the recognized negative correlation between agricultural production and services such as water quality, carbon storage, natural aesthetics, soil organic carbon, and habitat protection [18,20,84]. The synergies between ES based on lakes transformed to trade-offs among ES dependent on hydraulic engineering. For reclamations and large-scale agricultural production, a continuous investment was required for the Sihu basin in water conservancy. Improved flood protection dikes, large drainage pumping stations, and more irrigation culverts were needed to deal with problems such as low-lying farmland, waterlogging and high levels of groundwater. The cost of this transition was huge and unsustainable for the lowland of the lake basin. Although flood control and drainage facilities had expanded the scope of agricultural production, people were exposed to unexpected huge risks under the increasingly extreme climate—the well-known levee effect [85]. The transition from lake-based ES supply to engineered services in Sihu agricultural watershed has important implications for emphasizing nature-based human modification [86,87].

## 5. Conclusions

This paper presented a spatiotemporal approach to mapping the ecosystem service supply and identified the effect of human modification on ecosystem services over the past seven decades. First, the incorporation of historical ecosystem services can influence the synergies and trade-offs between pairs of ecosystem services. The spatial snapshot correlations among ES gave rise to misunderstanding (such as water yield and runoff control) and

missed synergies (such as water purification and recreational potential). Second, ecosystem service bundles showed that most ecosystem services can be synergetic in one bundle, with multifunctionality before intensive modification, but later transformed to single or limited services dominated bundles, especially in lake-polder areas, which has important implications for spatial planning and priority settings. Third, the lake reclamation and hydraulic infrastructure were the most significant modification indicators for explaining the variation in ecosystem services. The change of dominant drivers also revealed the transition from lake-based ecosystem service supply to engineered services. In other words, the relationship between humans and nature in Sihu decoupled during the process of modification. Overall, an improved understanding of the spatiotemporal pattern of historical ecosystem services and the underlying human modification influence is vital for realizing the multifunctionality of agricultural watershed and informing spatial planning.

Further research could be carried out based on this paper's framework. Firstly, the provision, distributions, and relationships of ES can differ across scales, while the drivers may act differently in their relationships with ES at various scales [88–90]. Assessing the spatiotemporal dynamics of ecosystem services driven by human modification at different spatial scales needs to be considered. Secondly, long-term primary data are often not available for many ES. For example, the pollination service is very important in agricultural watershed [91,92], which is ignored in this research because of the data availability. The cultural ES are also consistently recognized but not integrated within the ES framework, except the recreation service [93]. The indicators of cultural ES, such as aesthetic, traditional knowledge, and sense of place should be developed. Finally, the mapping of ES and ES bundles can be used for many spatial issues, such as conservation planning [33], land capability classification and suitability assessment in agricultural watershed [94], and priority setting in watershed ecological restoration [95]. These future research directions will increase the realism and policy relevance of ecosystem services assessment.

**Supplementary Materials:** The following supporting information can be downloaded at: https://www.mdpi.com/article/10.3390/land12030577/s1 [96–105].

**Author Contributions:** Conceptualization, H.L. and H.Y.; methodology, H.L.; software, H.L.; formal analysis, H.L.; writing—original draft preparation, H.L.; writing—review and editing, H.L. and H.Y.; supervision, H.Y. All authors have read and agreed to the published version of the manuscript.

**Funding:** This research was founded by the National Key Research and Development Project of China (Grant No: 2016YFC0401108) and the National Natural Science Foundation of China (Grant No: 51678002).

**Data Availability Statement:** Not applicable.

**Acknowledgments:** We would like to thank all editors and reviewers for their insightful comments, which helped us improve the quality of this paper.

**Conflicts of Interest:** The authors declare no conflict of interest.

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
