# Peer review of "Spatiotemporal Dynamics of Ecosystem Services Driven by Human Modification over the Past Seven Decades: A Case Study of Sihu Agricultural Watershed, China"

_land, doi:10.3390/land12030577_

Round 1

Reviewer 1 Report

Title: The Spatiotemporal Dynamics of Ecosystem Services Driven by Human Modification over the Past Seven Decades: A Case Study of Sihu Agricultural Watershed

1) Observations:

- Line 10 to 11 "The historical dynamics of ecosystem services are important in detecting the impacts before and after intensive modification, but are discussed rarely": This information is not truthful in many countries where its studies happen.

- Line 119: DEM is altimetry in meters. The spatial resolution needs to be improved. Figure 1 should also represent the water network if the area is a watershed.

- Line 143: Figure 2 could show only one legend and scale (Both are the same for all maps) and improve the image resolution.

- Line 127 to 128 "Five classes of land use types (settlements, lakes, marsh and herbaceous wetlands, reservoirs and fish ponds, and cultivated land): Figure 2 presents five classes - wetlands, sediment, pond, lake, and agriculture agriculture. The author needs to describe only one class denomination form.

- Line 311 to 312: Figure 2 could have the same color scale. All maps have only low and high references, without number definitions.

- Figure 4 (Line 317) and Line 156 (RE: rice; CN: cotton; OC: oil crop; FP: freshwater products): Is it described as ecosystem services? Or it's evaluated considering the ability to supply ecosystem services?

- Line 360 How (Population density, hydraulic engineering intensity, lake reclamation intensity, fertilizer and insecticide input were driving forces that could explain 40% to 60% of the distribution and change of ES in the Sihu basin): How were identified or measured?

2) Final considerations:

- The article presents a methodology relevant based on the multicriteria evaluation.

- The main observation is the concept of ecosystem services adopted, if the correlations are between the agriculture-freshwater (rice, cotton, oil crop) products and the other factors (water yield, runoff, habitat quality, total phosphorus export, total nitrogen export, carbon storage, soil organic matter, ph value, recreation potential), or if all of them were used as ecosystem services.

[Line 32: Ecosystem services (ES) are “benefits acquired by human from the ecosystem”, including material goods and services, which are closely bound up with human well- being.]

[Line 280 to 282: “Water purification services in the basin generally decrease with rising grain products. In the past two decades, a trade-off was shown in the supply of aquatic products with various regulatory and cultural services, especially soil organic matter and habitat quality”.]

These points induce us to believe that only the natural environment is responsible for the service's answer.

- The item “Relationships between ES and human modification drivers” describe human dynamics, but it is not understandable without a previous historical retrospect or database presentation.

Reviewer 2 Report

The paper attempts to find the effects of human modification of ecological services in the Sihu Agricultural Watershed. This can be improved before acceptance and the following suggestions can be considered.

Title: "The" Spatiotemporal Dynamics of Ecosystem Services......."The" may be omitted. Write China or Zone after Sihu Agricultural Watershed.

Line 9....."human modification on ecosystem services interaction".....Revise the line.

In abstract: Results with data are missing. Information about soil carbon sequestration, crop, recreation (cultural services), etc. will attract the readers.

Keywords: Some more can be added highlighting the novelty of the study like habitat quality; soil quality......Write human modification drivers instead of human modification.

Line 95-97.....The annual mean maximum and minimum temperature can be provided. Give space after 1350.....1350mm 

Figure 1....The quality of image showing maps of China is not clear.

Line 166.....How was surface runoff measured? 

Line 218.....Use of subscript....CO2

How was soil organic carbon estimated? Similarly in case of nitrogen and phosphorus......Was the data procured from somewhere?

Line 245, 343....trade-off

Line 252....Check bracket......ggplot2() package in R

Line 256.....SOM is already elaborated in Line 56....Do not elaborate....Write SOM only....Revise the line.

The correlation results should be elaborated scientifically. Results are not well described. The authors should check the entire results section. Data and values are not explained.

Figure 4....Check image quality.

Line 359.....Redundant analysis or Redundancy analysis?

Line 359-380.....Redundancy analysis is not well explained as per the year.

The discussion section can be much improved

Line 439, 473....Check spacing before bracket...results[23,27].....modification[78,79].....Check such typos throughout the manuscript [Line 55, 56, 72, 77, 109, 112, 147-151].

Line 452....Different format followed.....(Rist et al., 2014)

Land capability classification and crop suitability could have been added in the article. However, these can also be included for future scope.......Such statement is expected in conclusion section as future scope. 

References: The starting letters in the name of Journals are small......science; International journal of environmental research and public health.....Check them. 

Reviewer 3 Report

The paper has attempted to highlight  an important issue in the real of environmental research citing case studies from their original research but preparing the manuscript needs some improvement 

The authors should improve the english in order to express their views transparently and avoiding mistakes in the spelling and grammers  Include some more relevent references in order to substantiate the write up with more scientific inputs.

Round 2

Reviewer 1 Report

The paper can be accept in the present form.